# Ethyl Pyruvate Promotes Proliferation of Regulatory T Cells by Increasing Glycolysis

**DOI:** 10.3390/molecules25184112

**Published:** 2020-09-09

**Authors:** Ivan Koprivica, Dragica Gajić, Nada Pejnović, Verica Paunović, Tamara Saksida, Ivana Stojanović

**Affiliations:** 1Department of Immunology, Institute for Biological Research “Siniša Stanković”—National Institute of Republic of Serbia, University of Belgrade, 11060 Belgrade, Serbia; ivan.koprivica@yahoo.com (I.K.); gajic_dragica@yahoo.com (D.G.); nada.pejnovic@gmail.com (N.P.); cvjetica@gmail.com (T.S.); 2Institute of Microbiology and Immunology, School of Medicine, University of Belgrade, 11000 Belgrade, Serbia; vericapaunovic@gmail.com

**Keywords:** regulatory T cells (Treg), ethyl pyruvate, glycolysis, immunoregulation

## Abstract

Ethyl pyruvate (EP), a stable form of pyruvate, has shown beneficial effects in animal models of shock, ischemia/reperfusion injury, and sepsis due to its potent anti-oxidant and anti-inflammatory properties. Our recent study demonstrated that EP application prevented the clinical manifestation of type 1 diabetes in mice by augmenting regulatory T cell (Treg) number and function. Our present study shows that EP increases Treg proliferation and suppressive function (perforin and IL-10 expression) during in vitro differentiation from conventional CD4^+^CD25^−^ T cells. Enhanced expansion of Treg after EP treatment correlated with increased ATP levels and relied on increased glycolysis. Inhibition of oxidative phosphorylation did not attenuate EP stimulatory effects, suggesting that this metabolic pathway was not mandatory for EP-driven Treg proliferation. Moreover, EP lowered the expression of carnitine palmitoyltransferase I, an enzyme involved in fatty acid oxidation. Further, the stimulatory effect of EP on Treg proliferation was not mediated through inhibition of the mTOR signaling pathway. When given in vivo either intraperitoneally or orally to healthy C57BL/6 mice, EP increased the number of Treg within the peritoneal cavity or gut-associated lymphoid tissue, respectively. In conclusion, EP promotes in vitro Treg proliferation through increased glycolysis and enhances Treg proliferation when administered in vivo.

## 1. Introduction

Ethyl pyruvate (EP) is an aliphatic ester composed of ethanol and pyruvate. It is commonly used in the food industry as an additive. EP easily enters the cell without the need for transporters, and can be degraded and converted to pyruvate. The obtained pyruvate may enter the tricarboxylic acid (TCA) cycle and, therefore, EP is considered as cellular metabolic fuel for the generation of energy [1,2]. In clinical practice, systemic administration of pyruvate is used as a therapeutic intervention for cardiac, neurological, and acid-base disorders [3]. However, pyruvate is unstable in aqueous solutions, and this certainly limits its therapeutic potential. There are data on the beneficial role of exogenously given EP, a more stable compound than pyruvate, in animal models of oxidant-mediated cellular injury [2]. The diverse pharmacological effects of EP include the inhibition of pro-inflammatory cytokines secretion, attenuation of reactive oxygen species (ROS)-mediated damage to cells and tissues, inhibition or promotion of apoptosis (depending on the circumstances), and boosting of cellular ATP synthesis. These activities of EP have been shown in numerous animal models of inflammatory diseases, including uveitis [4], sepsis [5], and ischemia/reperfusion injury [6]. In general, the effects of EP are attributed to the inhibition of high mobility group box 1 (HMGB1), an alarmin that signals cell damage. Further, EP acts as a potent inhibitor of nuclear factor-κB (NF-κB), which is commonly considered as a transcription factor for pro-inflammatory cytokines [2,7]. Our previous study [8] demonstrated that during the development of type 1 diabetes in mice, EP skewed the immune response towards anti-inflammatory, i.e., in vivo administration of EP favored tolerogenic dendritic cells and regulatory T cells (Treg) in the pancreatic lymph nodes and pancreatic infiltrates. Moreover, we showed that EP stimulated Treg proliferation and migration to the inflamed tissue and increased Treg suppressive properties in an animal model of type 1 diabetes [8].

Treg have the potency for the prevention of autoimmunity and strong immunosuppression. Their energy requirements and metabolic processes are quite different compared to T helper (Th) cells. It is usually stated that Treg have low expression of glucose receptors and mainly rely on fatty acid oxidation (FAO) and subsequent oxidative phosphorylation (OXPHOS) [9]. However, Treg proliferation demands additional energy provided by enhanced glycolysis [10]. Recent studies confirm that the glycolytic pathway is necessary for Treg activation as human Treg cultured ex vivo in the presence of the anti-CD3 antibody, irradiated antigen-presenting cells, and IL-2, showed elevated glycolysis in comparison to the conventional T cells [11]. In addition, when cultured in vitro, Treg engage both glycolysis and FAO in contrast to the conventional T cells that utilize only glycolysis [12]. During OXPHOS, ROS is generated, and it has been found that the total ROS concentration in Treg is significantly greater in comparison to the other T cell subsets both in mice and humans [13,14,15]. ROS is actually needed for the differentiation of naïve conventional CD4^+^ cells into induced Treg (iTreg) [16,17]. Therefore, substances that can alter immune cell metabolism may affect Treg differentiation, proliferation, and suppressive function. The exact mechanism of the EP-mediated stimulatory effect on Treg has not yet been elucidated.

In this study, we aimed to investigate the mechanisms of EP-dependent effects on Treg in relation to cell metabolism. In addition, we explored the effects of in vivo administered EP on Treg in the peritoneum and gut-associated lymphoid tissue. The obtained results indicated that EP acts as metabolic fuel for differentiating Treg by increasing glycolysis, thus enhancing Treg proliferative capacity.

## 2. Results

### 2.1. EP Stimulates the Proliferation of In Vitro Differentiated Treg

In order to determine the timeline of EP effect on Treg differentiation in vitro, CD4^+^CD25^−^ cells isolated from spleens of C57BL/6 mice were stimulated with the conventional Treg differentiation cocktail that is comprised of plate-bound anti-CD3 and soluble anti-CD28 antibodies together with IL-2 and TGF-β. At the same time, EP was applied 24 h after the beginning of cell culture. EP increased the proportion of Treg even 24 h after application, while the maximal increase was observed 72 h after EP addition (Figure 1A). The increased number of Treg after EP treatment most likely resulted from the proliferation of already differentiated Treg, rather than from the possible direct impact on Treg differentiation, as determined by carboxyfluorescein succinimidyl ester (CFSE) dilution assay 48 h after EP application (Figure 1B). EP also increased the proportion of Treg that produced perforin, while granzyme production and CD39 (an ATP ecto-enzyme) expression remained unchanged (Figure 1C). The expression of IL-10 mRNA was also increased in the presence of EP, while there was no difference in the expression of TGF-β mRNA between conventionally and EP-induced Treg (Figure 1D).

### 2.2. EP Increases Energy Production in Treg during Differentiation

Along with increased Treg proliferation, EP stimulated cellular energy production, as evidenced by the increased intracellular content of ATP (Figure 2A). Therefore, we wanted to investigate which step of the metabolic cascade was affected by EP: glycolysis, TCA, FAO, or OXPHOS. Seemingly, EP-driven Treg proliferation relied on the functional glycolytic pathway, as the application of 2-deoxyglucose (2-DG), an inhibitor of glycolysis, inhibited the observed increase in Treg proportion (Figure 2B). Furthermore, Treg in the presence of EP expressed more hexokinase 2 (HK2), a key regulatory enzyme for glycolysis that catalyzes the initial step of glucose phosphorylation to produce glucose-6-phosphate (Figure 2C). EP increased the mRNA expression for glyceraldehyde-3-phosphate dehydrogenase (GAPDH), which catalyzes the conversion of glyceraldehyde 3-phosphate to d-glycerate 1,3-bisphosphate during glycolysis (Figure 2D). In addition, EP increased the expression of hypoxia-inducible factor 1-α (HIF-1α), a glycolysis stimulator, compared to conventionally-induced Treg (Figure 2E).

To investigate the role of EP in the conversion of pyruvate to acetyl-CoA (a precursor for TCA cycle), we determined the protein expression levels of pyruvate dehydrogenase (PDH) and kinase of pyruvate dehydrogenase 4 (PDK4), which serves as a negative regulator of PDH activity. PDH was similarly activated (Figure 2F), and PDK4 was similarly expressed (Figure 2G) in both conventionally-induced and EP-treated Treg.

To test the possibility that EP modulates FAO in differentiating Treg, we evaluated the expression of carnitine palmitoyltransferase I (CPT1), which is involved in the conversion of fatty acyl CoA into fatty acyl-carnitine, an essential step in FAO. Results suggest that CPT1 was significantly down-regulated after EP treatment (Figure 2H).

To explore the possible influence of EP on the OXPHOS pathway, we determined the expression of mitochondrial inner membrane electron transfer complexes, which remained the same as in conventionally-induced Treg (Figure 3A). Interestingly, the application of rotenone, an inhibitor of the mitochondrial Complex II, did not alter Treg differentiation in both conventionally and EP-treated cells (Figure 3B). However, it did inhibit the production of ROS (Figure 3C), suggesting that the OXPHOS pathway is redundant for in vitro Treg differentiation. Since EP was found to elevate the proportion of ROS^+^ Treg 24 h after it was added to the cell culture (Figure 3C), we further determined the timeline of ROS production. ROS production, both in terms of the number of ROS-producing cells and the level of ROS production per cell, did not immediately change following EP administration (after 1 or 3 h) (Figure 3D,E). However, 18 h after the addition of EP, the proportion of ROS^+^ Treg and ROS content were elevated (Figure 3D,E), indicating that Treg differentiation is accompanied by an increase in their ROS production.

### 2.3. The Effect of EP on the mTOR Pathway in Differentiating Treg

Since the inhibition of the mTOR pathway is known to favor Treg differentiation, we compared the effects of EP and the mTOR inhibitor rapamycin. Their impacts on Treg differentiation were similar in regard to the Treg proportion increment (Figure 4A) and their ROS production (Figure 4B). However, when applied together, rapamycin and EP did not exert an additive stimulatory effect on Treg proportion (Figure 4A,B), suggesting that EP might affect a similar signaling pathway as rapamycin. In order to examine this, we determined the phosphorylation states of S6 kinase (on T389, downstream from the mTORC1 complex) and Akt kinase (on Ser423, downstream from the mTORC2 complex). As expected, rapamycin completely attenuated S6 kinase phosphorylation, while EP had no effect (Figure 4C), suggesting that the Treg-stimulatory effect of EP was not mediated by interference with the mTOR signaling pathway. While rapamycin and EP had no effect on the phosphorylation of Akt kinase on their own, when applied together, they significantly increased its phosphorylation (Figure 4D).

### 2.4. In Vivo Application of EP Increases Treg Numbers in the Peritoneum and the Gut

In order to investigate whether EP has the capacity to stimulate Treg in vivo in non-pathological settings, EP was given intraperitoneally or orally to healthy C57BL/6 mice. The intraperitoneal application resulted in an increase of the proportion (Figure 5A) and number (Figure 5B) of Treg in the peritoneal cavity after only two administrations of EP. Oral application during seven days increased the proportion (Figure 5A) and number (Figure 5B) of Treg in both lamina propria and the Peyer’s patches of C57BL/6 mice. In the peritoneum, the increased number of Treg after EP application resulted from the proliferation of Treg as determined by 5-bromo-2′-deoxyuridine (BrdU) incorporation into the dividing cells (Figure 6).

## 3. Discussion

This study shows that EP enhances in vitro Treg proliferation during the differentiation from conventional CD4^+^ T cells by acting as a stimulator of glycolysis. Further, EP shows the capacity to increase the number of Treg in vivo when applied intraperitoneally or orally to healthy mice.

As we have previously shown, EP can act as an anti-inflammatory agent by inhibiting the action of effector T cells and by promoting the regulatory phenotype of dendritic and T cells [8,18]. EP enhanced Treg proportions and their suppressive ability when applied simultaneously with streptozotocin-induced type 1 diabetes in C57BL/6 mice. In addition, our results indicated that EP could partially replace the activity of TGF-β during Treg differentiation in vitro [8]. Currently, we have shown that EP stimulates Treg proliferation in vitro rather than differentiation. This is based on the increased number of divided Treg after the addition of EP, while the number of undivided Treg was comparable to Treg differentiated in the presence of conventional stimulation. The effect of EP seems to be specific for Treg, since it did not affect the number of Th1 and Th17 during differentiation from conventional CD4^+^ T cells [8]. Similarly to the in vitro effect, EP applied in vivo through the oral route was stable, reached the intestine, and increased the proportion of Treg in both the lamina propria and the Peyer’s patches. Additionally, short-term application significantly increased the Treg proportion and their proliferation in the peritoneum, suggesting that EP may be a potent stimulator of Treg expansion even in healthy conditions.

Although it was anticipated that EP would act as a substrate for pyruvate generation in the cell and would subsequently fuel the TCA cycle, the results of this study suggest that EP potentiates the glycolytic cycle in proliferating Treg. The general opinion is that the metabolic pattern of effector T cells is predominantly glycolytic [19,20], while induced Treg mainly utilizes a distinct metabolic program based on mitochondrial oxidation of lipids and pyruvate [9,21,22,23]. However, recent studies have highlighted the necessity of glycolysis in Treg activation and proliferation [11,12]. Our results also imply that EP stimulated glycolysis, which is consistent with the observed increase in Treg proliferation after EP administration. It seems that the effect of EP was mediated through enhanced expression of glycolytic enzymes, HK2 and GAPDH. Similar effects of EP were observed in renal tubular endothelial cells where EP prevented hydrogen peroxide-induced reduction of hexokinase activity and ATP [24].

2-DG attenuated the EP-potentiated Treg number, which further confirms the role of glycolysis in EP-driven Treg proliferation. EP-treated cells produced more HIF-1α, which is known to stimulate the expression of most glycolytic enzymes and glucose transporters such as GLUT1 [10]. EP was indeed shown to have the ability to increase HIF-1α production and glycolysis in a cancer cell line, which further substantiates our findings [25]. Interestingly, these EP-driven Treg resemble thymus-originated fully formed Treg that rely on glycolysis for energy production [26]. Additionally, recent data suggest that in vitro differentiation of Treg depends on both glycolysis and FAO [12]. Our results with EP contradict the findings made by Chakhtoura et al., 2019 [27], where EP application down-regulated glycolysis in dendritic cells. In spite of this, it is firmly established that pyruvate conversion to lactate transforms NADH into NAD+, thereby maintaining the large NAD+/NADH ratio necessary for driving glycolysis [28]. Since EP can easily convert to pyruvate, the observed enhanced glycolysis could be the result of the above-described process. In addition, the proliferating capacity of dendritic cells and Treg is different, and EP effects may depend upon the activation status of the particular cell. For example, according to reference [27], EP inhibited glycolysis when applied to differentiated dendritic cells that are mainly quiescent. In contrast, in this study EP stimulated glycolysis but was applied to proliferating Treg suggesting that EP may engage different pathways depending on the cell type and their division status.

Treg basal levels of ROS are significantly higher compared to other T cell subsets [21] and Treg seem to be quite resistant to oxidative stress [17]. However, all T cells when activated through the TCR, generally upregulate ROS formation [29]. The observed increase in ROS production after EP treatment coincides with the increased Treg proportion. However, ROS is not relevant for the observed EP-stimulated Treg proliferation because the inhibition of OXPHOS by rotenone did not change the number of Treg. The observed ROS up-regulation after EP treatment was not immediate, but significantly postponed, and was probably a reflection of EP-driven enhancement of the Treg number. However, these higher levels of ROS might also serve as a stabilizer of FoxP3 expression, as the optimal nuclear factor of activated T-cells (NFAT) activity (transcription factor for IL-2) is ROS-dependent and induced by Ca^2+^ mobilization and mitochondrial metabolism [30]. Although the detected rise in ROS is contrary to the generally considered EP ROS scavenging capabilities, EP was previously shown to be able to stimulate ROS production through glycolysis stimulation, at least in cancer cells [25]. Increased ROS was not a consequence of mitochondrial multiplication, since the expression of electron transport complexes was the same in both EP-treated and conventionally-induced Treg. It is likely that ROS was generated through OXPHOS, as the inhibition of the electron transport chain by rotenone reduced ROS formation in both EP- and conventionally-induced Treg.

The observed effect of EP on Treg proliferation can be in direct relationship to the inhibition of FAO. During proliferation, the need for lipids and fatty acids as a part of cell membranes rapidly increases and, therefore, fatty acid synthesis is favored [31]. Therefore, the down-regulation of CPT1, an enzyme involved in FAO, could be related to the EP-forced Treg proliferation.

Treg suppressive capacity depends upon their production of versatile inhibitory mediators and the inhibition of T cell activation by cell-to-cell contact [32]. We have shown that in vitro EP stimulates the expression of CTLA-4, PD-1 and GITR [8], as well as IL-10 and perforin expression. These results indicate that EP-treated Treg possess an increased suppressive capacity, as IL-10 is known to down-regulate Th1 and Th17 activity. In addition, the increased presence of perforin, which enables membrane pore formation in the target cells, could be the pathway utilized by EP-treated Treg for their suppressive activity. Finally, the elevated release of ROS from Treg could be detrimental for effector T cells [33,34].

EP aids Treg proliferation without promoting mTOR inhibition. Generally, mTOR inhibition serves as a positive signal for FoxP3 expression [35]. As both rapamycin and EP had similar effects on increasing Treg proportions in vitro, and their joint application had no additive effect, we hypothesized that EP interferes with the mTOR pathway as well. While rapamycin completely abrogated S6 kinase phosphorylation (downstream of mTORC1), EP-treated Treg had a similar phosphorylation pattern as conventionally-differentiated Treg, suggesting that EP was not involved in mTOR inhibition during the promotion of Treg proliferation. The obviously present mTORC1 activity correlates well with the established glycolysis necessity for conventional or EP-driven Treg differentiation. This is in accordance with the notion that mTORC1 activity is required to sustain high levels of aerobic glycolysis [36,37]. While this is usually the case in effector CD4^+^ and CD8^+^ T cells, in this in vitro Treg differentiation protocol glycolysis seems to be the predominant energy source needed for Treg differentiation and proliferation.

In conclusion, EP promotes Treg proliferation in vitro by enhancing the glycolytic pathway and energy production. In vivo, Treg proportion is also increased by either short-term or long-term application of EP in healthy animals. Our results imply that the concept of a strict dichotomy between glycolysis and oxidative phosphorylation as energy sources for effector T cells and Treg, respectively, may be over-simplistic and, therefore, targeting the glycolytic pathway for Treg expansion and suppressive function by EP could be beneficial. Thus far, EP has been shown to be safe when tested in both healthy human volunteers and heart surgery patients [2,38]. Our data suggest that the utilization of EP may be a promising supporting therapy for autoimmune and immunoinflammatory diseases in which the enhancement of Treg numbers and suppressive functions is expected to dampen disease activity and progression.

## 4. Materials and Methods

### 4.1. Mice

C57BL/6 mice maintained at the animal facility at the Institute for Biological Research “Siniša Stanković”, under standard conditions. Internal Ethics Committee approved all experimental procedures (App. No. 01-11/17-01-2475) in accordance with the Directive 2010/63/EU. In addition, experiments in mice conformed to ARRIVE guidelines.

### 4.2. In Vivo EP Application

EP was administered intraperitoneally (100 mg/kg bw) to healthy male C57BL/6 mice 2 times with 6 h apart. Control mice were treated with an equal volume of Hartmann’s solution (Hemofarm A.D., Vršac, Serbia) as a vehicle. Peritoneal cells were isolated 18 h later and stained for Treg. For the determination of Treg in gut-associated lymphoid tissue, EP or vehicle were administered orally to healthy male C57BL/6 mice for 7 days (100 mg/kg bw), and on the 8th day, mononuclear cells from the lamina propria and Peyer’s patches were isolated.

### 4.3. In Vivo Proliferation Assay

EP was administered intraperitoneally (100 mg/kg bw) to healthy male C57BL/6 mice for 2 days, twice each day, with 6 h apart. Control mice were treated with an equal volume of Hartmann’s solution (Hemofarm A.D., Vršac, Serbia) as a vehicle. Peritoneal cells were isolated 18 h later and stained for Treg. For measurement of lymphocyte cell proliferation, 5-bromo-2′-deoxyuridine (BrdU, 100 mg/kg bw) (Sigma-Aldrich, St. Louis, MO, USA) was applied intraperitoneally to mice 24 h before analysis.

### 4.4. Treg Differentiation Protocol

CD4^+^CD25^−^ conventional T cells were obtained from spleens of healthy C57BL/6 mice. Organs were removed aseptically, dispersed through a cell strainer (70 μm) and cell suspension was spun at 500 g for 5 min. Red Blood Cell lysis buffer (Thermo Fisher Scientific, Waltham, MA, USA), was used for erythrocyte lysis. Spleen cells were then washed with PBS containing 3% fetal calf serum (FCS). The obtained splenocytes were first incubated with a biotin anti-mouse CD25 antibody for 20 min on ice. The Cells were then resuspended in cold magnetic bead buffer (PBS 0.5% BSA, 2 mM EDTA) and incubated with BD IMag™ Streptavidin Particles Plus–DM (BD Biosciences, Bedford, MA, USA) for 30 min on ice. Negatively selected cells were obtained after placing the tubes in BD IMag™ Cell Separation Magnet (BD Biosciences) for 8 min (2 repeats). These cells were then incubated with biotin anti-mouse CD4 antibody diluted in PBS with 3% FCS for 20 min on ice. Both anti-CD4 and anti-CD25 antibodies were from eBioscience, San Diego, CA, USA. The next step of magnetic separation was the same as for negatively selected cells. The cells were finally resuspended in a T lymphocyte medium—RPMI supplemented with 10% FCS, 1% penicillin and streptomycin, 5 μM β-mercaptoethanol, 0.02 mM Na-pyruvate, 25 mM HEPES, and 2 mM L-glutamine.

For conventional Treg differentiation, the obtained CD4^+^CD25^−^ cells (5 × 10^5^ cells per well) were seeded in U-bottom 96-well plate in T lymphocyte medium and stimulated recombinant mouse TGF-β (2 ng/mL) and IL-2 (10 ng/mL) (both from R&D Systems, Minneapolis, MN, USA), soluble anti-mouse CD28 antibody (1 μg/mL) and plate-bound anti-mouse CD3 antibody (1 μg/mL) (both from eBioscience). EP (125 μM), rotenone (5 nM), rapamycin (100 ng/mL), or 2-Deoxy-d-glucose (2-DG, 40 μg/mL) (all from Sigma-Aldrich) were added 24 h after the initiation of Treg differentiation. The phenotype of differentiated Treg was determined 72 h (or 48 h) after the addition of the specified substances, while ROS and ATP were determined at the indicated time points.

### 4.5. Ex Vivo Cell Isolation

For the isolation of lamina propria (LP) cells, the small intestine was freed from its contents and the Peyer’s patches, cut open longitudinally, then cut into 1 cm pieces and washed 3 times in PBS. The tissue pieces were then washed 3 times with PBS containing 10% FCS, 5mM EDTA on an orbital shaker (350 RPM, RT) for 15 min. After a final 5 min wash in RPMI 10% FCS, the samples were incubated with RPMI 10% FCS, collagenase D (500 μg/mL, Sigma-Aldrich) on an orbital shaker (500 RPM, 37 °C) for 1 h. The reaction was stopped with PBS 10% FCS, 5 mM EDTA, and the digested cell suspensions with the remaining tissue pieces were passed through a cell strainer. The obtained samples were washed twice with PBS 10% FCS, 5 mM EDTA, and the lamina propria mononuclear cells were finally isolated from the interface between 40% and 80% discontinuous Percoll gradient, after centrifugation at 2000 RPM for 20 min. For the isolation of immune cells from Peyer’s patches, the Peyer’s patches were passed through a cell strainer in PBS 3% FCS. For the isolation of peritoneal immune cells, peritoneal lavage was performed with 4 mL of cold PBS. After centrifugation, all cells were finally resuspended in RPMI 5% FCS.

### 4.6. Flow Cytometry

Surface molecules were detected on viable cells by using the following antibodies: Anti-mouse CD4-FITC (rat IgG2b, κ), CD4 PerCP-Cyanine5.5 (rat IgG2a, κ), CD25-PE (rat IgG1, λ), CD39-PE (rat IgG2b, κ) (all from eBioscience). The staining was performed for 40 min at 4 °C in PBS 1% BSA. For intracellular cytokine staining, cells were first stimulated with Cell Stimulation Cocktail (plus protein transport inhibitors) (eBioscience) for 4 h at 37 °C. Cells were then fixed in 2% paraformaldehyde, permeabilized with a permeabilization buffer (eBioscience) and stained with anti-mouse IL-10-FITC (rat IgG2b, κ) (eBioscience). For intranuclear staining, the cells were permeabilized using the protocol and buffers from the Mouse Regulatory T cell Staining Kit (eBioscience) and stained with the following antibodies: Anti-mouse FoxP3-PE-Cyanine5 (rat IgG2a, κ), Perforin-PE (rat IgG2a, κ), Granzyme B-FITC (rat IgG2a, κ), BrdU-PE (mouse IgG1, κ) (all from eBioscience), Ki67-FITC (goat polyclonal antibody) (SantaCruz Biotechnology, San Diego, USA). For BrdU staining, an additional step was included before the antibody incubation: Denaturation in 2 N HCI containing 0.5% Triton X-100 for 30 min, and then neutralization with 0.1 M Na_2_B_4_O_7_. Isotype-matched controls were included in all experiments (eBioscience). Cells were analyzed on PartecCyFlow Space (Partec, Görlitz, Germany) by FlowMax software. Cells were first gated on live cells (empirically determined) and further gated adequately for the required analysis.

### 4.7. Measurement of Intracellular Reactive Oxygen Species

Dihydrorhodamine 123 (DHR) was used to detect reactive oxygen species (ROS). After isolation from the cell culture, cells were exposed to DHR (5 μM) for 20 min at RT. After washing, the cells were analyzed by flow cytometry. The mean fluorescence intensity (MFI) was determined as a measure of intracellular production of ROS.

### 4.8. ATP Bioluminescence Assay

For quantitative measurement of ATP production, the protocol and buffers from the ATP Bioluminescence Assay Kit HS II (Roche Diagnostics GmbH, Mannheim, Germany) were used. After the cells were isolated from the cell culture, they were resuspended in the Dilution buffer. An equal amount of Cell lysis reagent was added, and the samples were then transferred into a black microtiter plate and incubated for 5 min at RT. The Luciferase reagent was then added, and bioluminescence measurement was performed right away by using Chameleon Plate Reader (Hidex, Turku, Finland). A standard curve created from known ATP concentrations was used to calculate the sample concentration of ATP.

### 4.9. Western Blot

In vitro cultured cells (5 × 10^6^) were lysed with a buffer containing 62.5 mM Tris–HCl (pH 6.8), 2% SDS, 50 mM DTT, 10% glycerol, with the Protease Inhibitor Cocktail (all from Sigma-Aldrich). All samples were boiled with the 4 × SDS sample loading buffer, except for those used to detect mitochondrial OXPHOS proteins, due to their sensitivity to heating. Sample electrophoresis was performed on 12% SDS–polyacrylamide gel. By using a semi-dry blotting system (Semi-Dry Transfer Unit, GE Healthcare, Buckinghamshire, England), the protein samples were electro-transferred from the gel onto polyvinylidene difluoride membranes. The membranes were then either blocked with PBST (PBS 0.1% Tween-20, Sigma-Aldrich) containing 5% BSA and probed with specific antibodies diluted in PBST 1% BSA, or blocked with TBST (TBS 0.1% Tween-20) 5% BSA and probed with specific antibodies diluted in TBST 5% BSA, depending on the manufacturer’s instructions. The secondary antibody for anti-mouse phospho-E1-α PDH (1:600), anti-mouse E1-α PDH (1:1000), anti-mouse PDK4 (1:1000) (all from Abcam), anti-mouse phospho-p70 S6K (1:600), anti-mouse p70 S6K (1:600), anti-mouse phospho-Akt (1:600), anti-mouse Akt (1:600) (all from Cell Signaling Technology, Danvers, MA, USA), anti-mouse HK2 (1:1000, Invitrogen) and anti-mouse CPT1 (1:800, SantaCruz Biotechnology) was HRP conjugated anti-rabbit IgG (1:3000, Cell Signaling Technology), while for Total OXPHOS Rodent WB Antibody Cocktail (1:500, Abcam) and anti-mouse β-actin (1:1000, Sigma-Aldrich) it was HRP conjugated anti-mouse IgG (1:5000, Invitrogen). Detection was achieved with Immobilon Western Chemiluminescent HRP Substrate (Millipore, Billerica, MA, USA), and the signal was captured with X-ray film (Kodak, Rochester, NY, USA). Densitometry was performed with Fiji, an open-source software for biological image analysis [39], and the production of specific proteins was presented relative to the production of either their non-phosphorylated protein forms or of β-actin.

### 4.10. Reverse Transcription and Real-Time PCR

TriReagent (Metabion, Martinsried, Germany) was used for dissolving the samples, and centrifugation with chloroform at 12,000 g was performed subsequently. After RNA isolation from the aqueous layer and precipitation with isopropanol, reverse transcription was performed. Samples (1 µg) were incubated with random hexamer primers and RevertAid™ M-MuLV Reverse Transcriptase (Fermentas, Vilnius, Lithuania). Target sequences of cDNA were amplified using SYBRGreen PCR master mix (Applied Biosystems, Woolston, UK) in Real-time PCR machine (Applied Biosystems). Primer pairs for IL-10 were 5′-TGTGAAAATAAGAGCAAGGCAGTG-3′ and 5′-CATTCATGGCCTTGTAGACACC-3′ (NM_010548.1), for TGF-β they were 5′-GACCCTGCCCCTATATTTGGA-3′ and 5′-CGCCCGGGTTGTGTTG-3′ (NM_011577.2), for GAPDH they were 5′-AGGTCGGTGTGAACGGATTTG-3′ and 5′-TGTAGACCATGTAGTTGAGGTCA-3′ (NM_001289726.1) and for β-actin they were 5′-GACCTGACAGACTACC-3′ and 5′-GGCATAGAGGTCTTTACGG-3′ (NM_007393.2). Gene expression was determined as 2^−(Ct-Ca)^, where Ct was the target gene cycle threshold, and Ca was the β-actin cycle threshold. SDS2.1 software (Applied Biosystems) was used to analyze the obtained cycle thresholds.

### 4.11. Statistical Analysis

Data were presented as mean ± SD. The presented results were representative of 3 repeated experiments with comparable results. A two-tailed Student’s t-test was used to determine the significance of differences between groups. Differences were considered statistically significant if *p* < 0.05. GraphPad Prism 5 software (GraphPad Software, Inc., La Jolla, CA, USA) was used for statistical analyses.

## Figures and Tables

**Figure 1 molecules-25-04112-f001:**
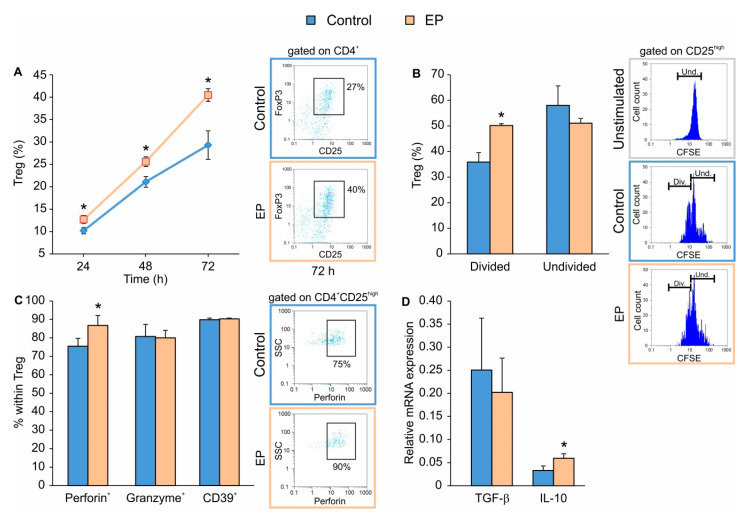
The effect of ethyl pyruvate (EP) on Treg in vitro. CD4^+^CD25^−^ T lymphocytes were stimulated for 24 h with a conventional Treg differentiation cocktail (control) and then additionally treated with 125 μM EP. (**A**) Treg (CD4^+^CD25^high^FoxP3^+^) proportion, at indicated time points after the addition of EP. Representative dot plots for the 72nd h are shown on the right-hand side. (**B**) Carboxyfluorescein succinimidyl ester (CFSE)-based proliferation of Treg (CD4^+^CD25^high^) was evaluated by flow cytometry (divided, Div. and undivided, Und. cells, with representative histogram plots shown on the right-hand side) 48 h after EP treatment. Unstimulated cells were stained with CFSE and served as the control for setting the threshold for undivided cells. (**C**) The proportion of Treg expressing perforin, granzyme or CD39, 72 h after EP treatment. Representative dot plots for perforin^+^ Treg are shown on the right-hand side. (**D**) mRNA expression of *Tgf-β* and *Il-10* in cultures treated with or without EP. * *p* < 0.05 represents the significant difference of EP-treated cells in comparison to control cells.

**Figure 2 molecules-25-04112-f002:**
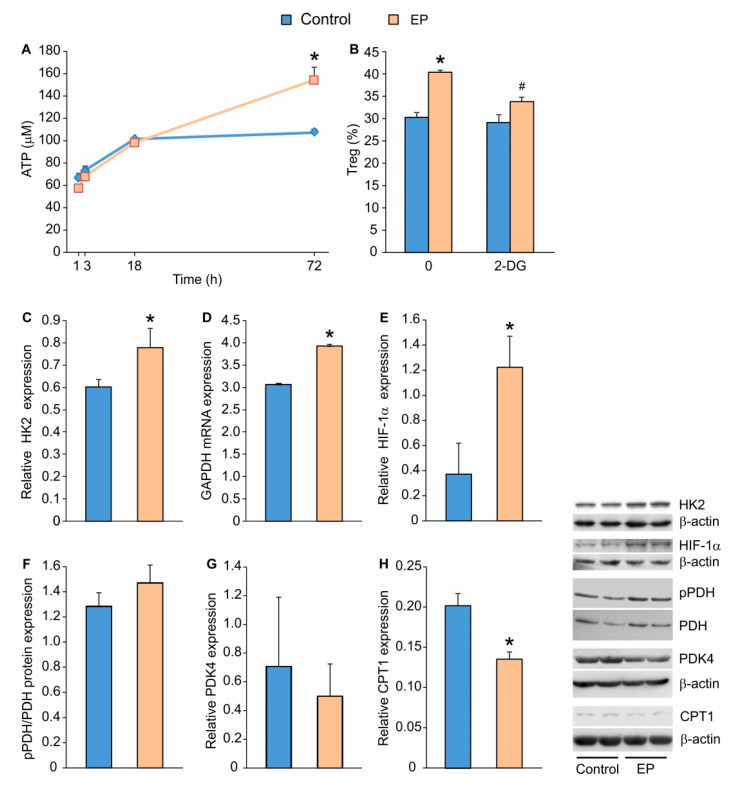
The influence of EP on metabolic pathways in differentiating Treg. (**A**) ATP cell content measured by chemiluminescence method in cells first exposed to a conventional Treg differentiating cocktail for 24 h and after EP treatment, at indicated time points. (**B**) The effect of 2-deoxyglucose (2-DG) on Treg (CD4^+^CD25^high^FoxP3^+^) proportion in control cells or EP-treated cells, measured 72 h after 2-DG addition. Relative protein expression (compared to β-actin or the total protein form) measured by immunoblot 72 h after EP administration: (**C**) HK2, (**E**) HIF-1α, (**F**) phosphorylated PDH (pPDH), (**G**) PDK4 and (**H**) CPT1. (**D**) Relative GAPDH mRNA expression (compared to β-actin) measured 72 h after EP administration. Representative blots are shown on the right-hand side. * *p* < 0.05 represents the significant difference of EP-treated cells in comparison to control cells, while # *p* < 0.05 represents the significant difference between 2-DG+EP-treated cells in comparison to EP-treated cells.

**Figure 3 molecules-25-04112-f003:**
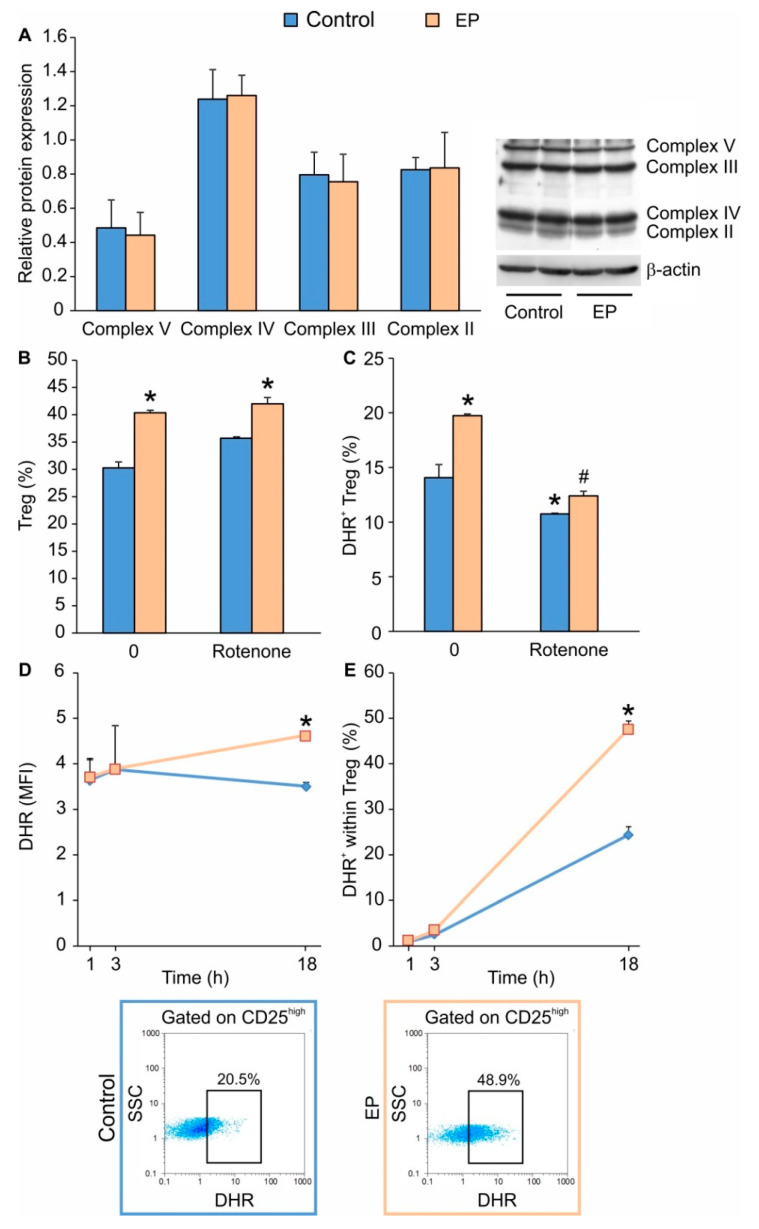
The effect of EP on oxidative phosphorylation (OXPHOS) and ROS production in differentiating Treg. (**A**) Relative protein expression of mitochondrial transport chain complexes 72 h after EP administration. The impact of rotenone (simultaneously applied with EP) on (**B**) EP-driven Treg (CD4^+^CD25^high^FoxP3^+^) proportion or (**C**) ROS formation, measured by dihydrorhodamine 123 (DHR) staining. (**D**) Timeline of ROS production per Treg (CD4^+^CD25^high^) and (**E**) the proportion of ROS-producing Treg in control and EP-treated cells. Representative dot plots are shown below the graph. * *p* < 0.05 represents the significant difference of EP- or rotenone-treated cells in comparison to untreated control cells, while # *p* < 0.05 represents the significant difference between rotenone+EP-treated cells in comparison to EP-treated cells.

**Figure 4 molecules-25-04112-f004:**
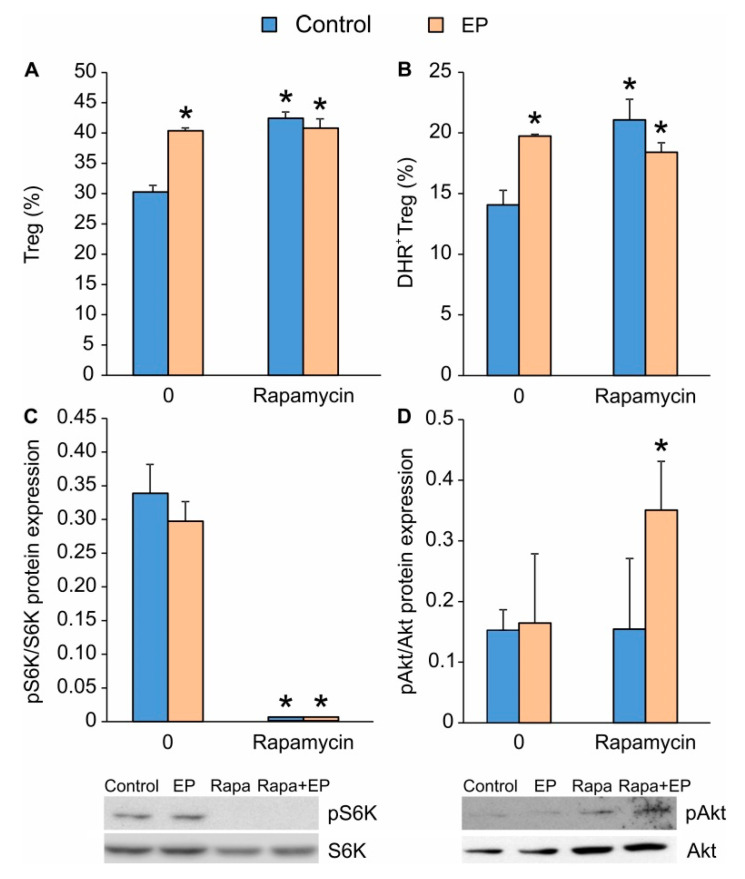
The influence of EP on mTOR activity. (**A**) The proportion of Treg (CD4^+^CD25^high^FoxP3^+^) and (**B**) the number of ROS-producing Treg (CD4^+^CD25^high^) 72 h after the treatment with EP and/or rapamycin (Rapa). (**C**) The expression of phosphorylated S6K (pS6K) and (**D**) phosphorylated Akt (pAkt) normalized to the expression of total protein forms measured by immunoblot 72 h after EP and/or rapamycin administration. Representative blots are shown below the graph. * *p* < 0.05 represents the significant difference of EP- or rapamycin-treated cells in comparison to untreated control cells.

**Figure 5 molecules-25-04112-f005:**
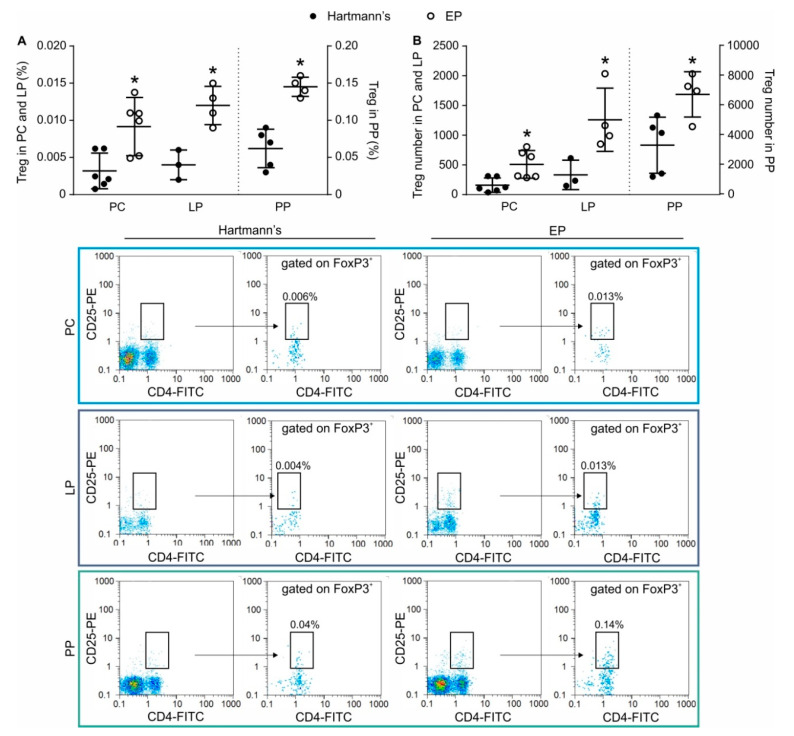
In Vivo EP influence on Treg numbers in the peritoneum and the gut immune system. EP was applied either by intraperitoneal injections or through the oral route to healthy C57BL/6 mice and (**A**) the proportion and (**B**) the number of Treg (CD4^+^CD25^high^FoxP3^+^) was determined ex vivo from the cell suspensions of peritoneal cells (PC), Peyer’s patches (PP), or lamina propria cells (LP). Representative dot plots and gating strategy are shown below the graph. * *p* < 0.05 represents the significant difference between EP-treated mice compared to vehicle-treated mice.

**Figure 6 molecules-25-04112-f006:**
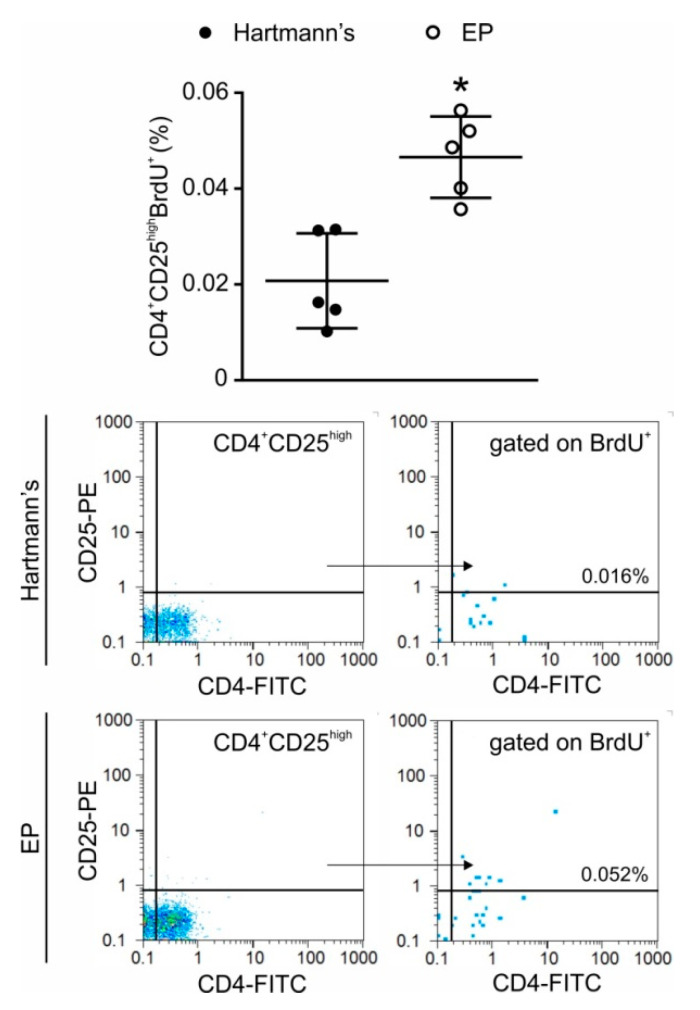
Treg proliferation after intraperitoneal EP application. EP was administered twice a day for two days, while BrdU was applied intraperitoneally 24 h before ex vivo analysis. Treg (CD4^+^CD25^high^) proliferation was determined by flow cytometry. Representative dot plots and gating strategy are shown below the graph. * *p* < 0.05 represents the significant difference between EP-treated mice compared to vehicle-treated mice.

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
