# Peer review of "Ethyl Pyruvate Promotes Proliferation of Regulatory T Cells by Increasing Glycolysis"

_molecules, 2020, doi:10.3390/molecules25184112_

Round 1

Reviewer 1 Report

Dear Editor,

This article by Ivan Koprivica et al entitled “Ethyl pyruvate promotes proliferation of regulatory T cells by increasing glycolysis” presents original data on the mechanism by EP promoting in vitro Treg proliferation and suppressive function through increased glycolysis during in vitro differentiation of CD4+CD25- T cells.

The paper is quite informative, expanding our knowledge in the domain of EP  effects on Treg in relation to cell metabolism. However, before being published the manuscript still needs some improvements:

My main criticism

  1. To establish the action of EP on the functional glycolysis pathway, the Author driven Treg proliferation with 2- deoxyglucose (2-DG) that inhibited the observed increase in Treg proportion. Also they observed that Treg in the presence of EP expressed more HIF-1α, which is a glycolysis stimulator. However, similar studies have evaluated some other elements, for instances  the expression or activation (phosphorylation) of glycolytic enzymes to corroborate the effect on the enhanced glycolysis pathway. Why did not the author do this analysis?  Are their results enough to establish that effect ?
  2. How could the Author explain the different effect observed in other study where EP inhibits glycolysis during dendritic cells activation phase?

Chakhtoura M., et al. Ethyl Pyruvate Modulates Murine Dendritic Cell Activation and Survival Through Their Immunometabolism. Front. Immunol., 28 January 2019 | https://doi.org/10.3389/fimmu.2019.00030

  1. At the figures 2A, 3D and 3E, the scale of time should be changed. The distance between points is not equal
  2. Verify the addition of text that do not correspond

Author Response

Reviewer 1

My main criticism

  1. To establish the action of EP on the functional glycolysis pathway, the Author driven Treg proliferation with 2- deoxyglucose (2-DG) that inhibited the observed increase in Treg proportion. Also they observed that Treg in the presence of EP expressed more HIF-1α, which is a glycolysis stimulator. However, similar studies have evaluated some other elements, for instances the expression or activation (phosphorylation) of glycolytic enzymes to corroborate the effect on the enhanced glycolysis pathway. Why did not the author do this analysis? Are their results enough to establish that effect?

We have determined the expression of hexokinase 2 protein and GAPDH mRNA and found increased expression of both enzymes necessary for glycolytic cycle progression (line 108-113 in the revised manuscript, Figure 2C and D). Therefore, we can now firmly state that EP stimulates glycolysis in Treg. We have also added literature data that corroborates our finding that EP affects hexokinase 2. Briefly, it was shown in renal tubular epithelial cells in vitro that EP mediates protection from hydrogen peroxide-induced reduction of hexokinase activity (Discussion, line 227-229).

  1. How could the Author explain the different effect observed in other study where EP inhibits glycolysis during dendritic cells activation phase?

Chakhtoura M., et al. Ethyl Pyruvate Modulates Murine Dendritic Cell Activation and Survival Through Their Immunometabolism. Front. Immunol., 28 January 2019 | https://doi.org/10.3389/fimmu.2019.00030

As the reviewer noticed our results with EP contradict the findings made by Chakhtoura et al., 2019 where EP application down-regulated glycolysis in dendritic cells. In spite of this, it is firmly established that pyruvate conversion to lactate transforms NADH into NAD+, thereby helping to maintain the large NAD+/NADH ratio necessary for driving glycolysis [Rogatzki MJ, Ferguson BS, Goodwin ML, Gladden LB. Lactate is always the end product of glycolysis. Front Neurosci. 2015;9:22. Published 2015 Feb 27. doi:10.3389/fnins.2015.00022.]. Since EP can easily convert to pyruvate, the observed enhanced glycolysis could be the result of the above illustrated process. In addition, the proliferating capacity of dendritic cells and Treg is different and EP effects may depend upon the activation status of the particular cell. For example, according to Chakhtoura et al., 2019, EP inhibited glycolysis when applied to differentiated dendritic cells that are mainly quiescent. In contrast, in this study EP stimulated glycolysis but was applied to proliferating Treg suggesting that EP may engage different pathways depending on the type of cell in question and their division status. We have included this paragraph in the revised version of the manuscript (Discussion, line 238-247).

  1. At the figures 2A, 3D and 3E, the scale of time should be changed. The distance between points is not equal.

We have changed the X axes in the indicated Figures 2 and 3.

  1. Verify the addition of text that do not correspond.

We have deleted non-corresponding text (line 77).

Reviewer 2 Report

The manuscript „Ethyl pyruvate promotes proliferation of regulatory T cells by increasing glycolysis“ (Koprivica I. et al.) represents an interesting and well-designed study, which characterizes the metabolic pathways by which ethyl pyruvate (EP) supports the proliferation and functions of regulatory T cell (Treg) in vitro and in vivo.

Using the adequate technology for the isolation, activation and phenotypic characterization of Treg cells (flow cytometry) and for the determination of involved metabolic pathways (ATP bioluminescence assay, measurement of intracellular reactive oxygen species (ROS), Western blotting for the detection of involved enzymes, HIF-1α and mTOR activity and reverse transcription or real-time PCR for the detection of IL-10 and TGF-β mRNA) the authors showed that EP given in vitro enhances Treg proliferation during their differentiation from conventional CD4+ T cells acting as a stimulator of glycolysis. The hypothesis was supported by findings that inhibition of oxidative phosphorylation did not attenuate EP effects, that stimulatory effect of EP on Treg proliferation was not mediated through inhibition of the mTOR signalling pathway, as well as that 2-deoxyglucose, an inhibitor of glycolysis, attenuated the EP-potentiated Treg number. Moreover, it was shown that EP-treated cells produced more ROS and HIF-1α, which is known to stimulate the expression of most glycolytic enzymes and that treatment with EP lowered the expression of carnitine palmitoyltransferase I, an enzyme involved in fatty acid oxidation. Besides, in additional experiments the authors showed that EP given in vivo either intraperitoneally or orally increased the number of Treg within the peritoneal cavity and in gut-associated lymphoid tissue (GALT) of healthy C57BL/6 mice.

The data are well presented and discussed. Mechanism and the type of induced Tregs remain to be elucidated, but the findings show that EP induce the metabolic reprogramming in life cycle of T cells and imply that EP may be used as a promising supporting therapy for autoimmune and immunoinflammatory diseases.  Importantly, the in vivo results presented in this study are in high agreements with reports showing that the GALT is the main site of extrathymic differentiation of Treg cells and that metabolites produced by commensal microorganisms during starch fermentation, such as short-chain fatty acids and butyrate, may facilitate extrathymic generation of Treg cells and contribute to development of tolerance to oral antigens.

Minor changes:

  • Line 15 .....  models of shock, ischemia/reperfusion injury and sepsis due to its
  • The abbreviation should be explained at the first appearance: Lines 34 (TCA), 41 (ROS), 45 (HMGB1), 46 (NF-κB), 82 (CFSE), 105 (HIF-1α), 142 (DHR).
  • Line 75 the sentence „The text continues here“ should be deleted
  • Line 76 It should be explained that the test cells were CD4+CD25- isolated from spleens of C57BL/6 mice
  • Line 103 add the underlined text   ....the application of 2-deoxyglucose (2-DG), an inhibitor of glycolysis inhibited the observed increase in Treg proportion 7
  • Line 179 should be A) the proportion and B) the number .....c

Author Response

Reviewer 2

We thank the reviewer for the useful comments. We have spell checked the entire manuscript and corrected grammar errors. 

Minor changes:

  1. Line 15 ..... models of shock, ischemia/reperfusion injury and sepsis due to its

We have changed the indicated sentence.

  1. The abbreviation should be explained at the first appearance: Lines 34 (TCA), 41 (ROS), 45 (HMGB1), 46 (NF-κB), 82 (CFSE), 105 (HIF-1α), 142 (DHR).

We have inserted the indicated abbreviations.

  1. Line 75 the sentence „The text continues here“ should be deleted.

The sentence is deleted.

  1. Line 76 It should be explained that the test cells were CD4+CD25- isolated from spleens of C57BL/6 mice.

We have included that the cells in question are CD4+CD25- isolated from spleens of C57BL/6 mice.

  1. Line 103 add the underlined text ....the application of 2-deoxyglucose (2-DG), an inhibitor of glycolysis inhibited the observed increase in Treg proportion.

We have added the proposed text.

  1. Line 179 should be A) the proportion and B) the number ....

We have corrected the text.

Round 2

Reviewer 1 Report

I agree with the changes and modifications made in the manuscript by the Authors

I recommend accepting in present form

Sincerely

PhD Leticia González Maya